# Differentiating between Monofloral Portuguese Bee Pollens Using Phenolic and Volatile Profiles and Their Impact on Bioactive Properties

**DOI:** 10.3390/molecules28227601

**Published:** 2023-11-15

**Authors:** Samar Larbi, Volkan Aylanc, Maria Shantal Rodríguez-Flores, Ricardo C. Calhelha, Lillian Barros, Feriel Rezouga, Maria Carmen Seijo, Soraia I. Falcão, Miguel Vilas-Boas

**Affiliations:** 1Centro de Investigação de Montanha (CIMO), Instituto Politécnico de Bragança, Campus de Santa Apolónia, 5300-253 Bragança, Portugal; samarlarbi@gmail.com (S.L.); volkan@ipb.pt (V.A.); calhelha@ipb.pt (R.C.C.); lillian@ipb.pt (L.B.); 2Laboratório Associado para a Sustentabilidade e Tecnologia em Regiões de Montanha (SusTEC), Instituto Politécnico de Bragança, Campus de Santa Apolónia, 5300-253 Bragança, Portugal; 3Département de Génies Biologique et Agroalimentaire, Université Libre de Tunis, 30 Avenue Kheireddine Pacha, Tunis 1002, Tunisia; feriel.rezouga@ult-tunisie.com; 4Departamento de Química e Bioquímica, Faculdade de Ciências, Universidade do Porto, 4169-007 Porto, Portugal; 5Facultad de Ciencias, Universidad de Vigo, Campus As Lagoas, 36310 Vigo, Spain; mariasharodriguez@uvigo.es (M.S.R.-F.); mcoello@uvigo.es (M.C.S.)

**Keywords:** bee products, phenolic composition, volatiles, antioxidants, cytotoxic activity

## Abstract

Nowadays, bee products are commended by consumers for their medicinal and dietary properties. This study aimed to differentiate between monofloral bee pollens originating from Portugal using phenolic and volatile profiles and investigate their antioxidant and cytotoxic activity. Total phenolic and flavonoid compounds were recorded between 2.9–35.8 mg GAE/g and 0.7–4.8 mg QE/g, respectively. The LC/DAD/ESI-MS^n^ analytical results allowed us to identify and quantify a total of 72 compounds, including phenolic and phenylamide compounds, whereas GC-MS results revealed the presence of 49 different compounds, mostly ketones, aldehydes, esters, hydrocarbons, and terpenes. The highest DPPH^•^ radical scavenging activity, EC_50_: 0.07 mg/mL, was recorded in the sample dominated by *Castanae* sp. pollen, whereas the *Rubus* sp. (1.59 mM Trolox/mg) and *Cistaceae* sp. (0.09 mg GAE/g) pollen species exhibited the highest antioxidant activity in ABTS^•+^ and reducing power assays, respectively. Regarding the anti-carcinogenic activity, only *Carduus* sp. showed remarkable cytotoxic potential against MCF-7.

## 1. Introduction

There is undeniable evidence that diet impacts human health, and dietary compounds are known to be one of the substantial therapeutic agents that can alter various cellular and molecular pathways [1]. Natural dietary compounds such as phenolics, vitamins, or volatile compounds act as a natural weapon against the more widespread diseases of today by reducing or preventing cellular oxidative stress [2,3,4]. Additionally, in vitro and in vivo studies, these compounds have also been demonstrated to be effective against ailments closely related to diet, including cancer, diabetes, and various inflammatory conditions [5,6,7].

From this perspective, bee pollen emerges as a natural dietary source, rich in both macro/micronutrients as well as bioactive compounds [8]. Even though bee pollen is an animal product, it is of plant origin. The bees collect the pollen and combine it with their own secretions, allowing these pollens to turn into a pellet structure, which is then transported to the hive by sticking to the pollen basket on the hind leg of the bees [8]. This final product is called bee pollen. The amount and diversity of nutritional and bioactive compounds in bee pollen varies depending on the plant(s) from which the pollen is collected [9,10]. Environmental factors such as the geographical region where the plant grows, the soil, and climate are also closely linked to the value of pollen [10]. Bee pollen is frequently evaluated based on whether it is mono or multifloral. Pollen that is predominantly obtained from a major taxon, comprising not less than 80% pollen from one flower, is called monofloral [11]. However, in most cases, bees collect pollen by visiting many flowers, and therefore, beekeepers obtain a mixture of pollen from various botanical origins when harvesting pollen, which is then classified as multifloral. In this case, a colour-based separation of bee pollen pellets to obtain monofloral bee pollen is necessary so that the biological activity or nutritional value of bee pollen with a fixed composition can be evaluated [12]. 

In Ancient Greece, Rome, and Egypt, bee pollen was viewed as a food source as well as a natural remedy [8]. Today, there is a great interest in researching and revealing the nutritional values and therapeutic properties of bee pollen. Efforts to study bee pollen from different botanical origins around the world have focused upon its detailed physicochemical composition [13], macro/micronutrients [14,15], and characterization of bioactive compounds [8,9,16] in order to ascertain what contributes to bee pollen’s anti-diabetes [17,18], antimicrobial [l,4], anticancer [6,19,20], and antioxidant biological properties [10,15,17]. However, the variability of the bee pollen composition, which is related to its botanical and geographical origin, represents a major drawback in the potential exploitation of this matrix.

A key aspect of setting bee pollen to an international standard, and determining its commercial value, is to evaluate the botanical source of the pollen and establish its physicochemical composition, bioactive compound composition, and biological activity [8,11,21]. Moreover, bee pollen from a single botanical source has a stable chemical composition, which makes it more applicable to different areas. There are few studies linking volatile organic composition in monofloral bee pollen with botanical origin. Additionally, most studies investigating the bioactive compounds of a natural product such as bee pollen have focused on phenolics, however, the bioactive compound composition of bee pollen, in most cases, is very rich in phenylamides [10]. There are unfulfilled findings concerning the presence of these compounds in bee pollen. Considering the above, this study aimed to evaluate the most representative monofloral bee pollen from Portugal, to reveal their bioactive composition, volatile composition, antiradical activity, and in vitro cytotoxic potential against several human tumour cell lines. Establishing links between bioactivity and botanical origin, illustrates the wide application potential for bee pollen in food and pharmaceutical fields.

## 2. Results and Discussion

### 2.1. Botanical Origin of Bee Pollen Samples 

Palynological analysis was performed after manually separating the bee pollen samples by colour, to elaborate on their botanical origins. Bee pollen is generally classified as monofloral in case it has a single major taxon at a relative frequency greater than 80% [11]. The results revealed that the majority of the samples, BP3 to BP7, showed a dominance of a single pollen species at 100%; the pollen species, *Carduus* sp. and *Ligustrum*/*Olea* sp., were dominant, at a rate higher than 80%, for the BP1 and BP2 samples, respectively (Table 1 and Figure 1). Although BP4 and BP6 samples were from different geographical locations (Nisa and Bragança, respectively), the same pollen type, *Rubus* sp., was dominant in both samples, with similar colours. This could be related to the fact that the *Rubus* sp. was dominant in the area where the pollen was collected by bees. Our results are similar to those previously reported for bee pollen samples in the north (Bragança) [10,12,22] and central regions (Nisa) of Portugal [23].

### 2.2. Total Phenolic and Flavonoid 

The colourimetric Folin–Ciocalteu and aluminium chloride assays, which are repeatable, easy, and reliable tests, are widely performed in accordance with phenolic and flavonoid standards, assuming equal responses from all phenolic and flavonoid compounds, especially when estimating the total amount of that bioactive compound’s classes in plant-based materials.

The total phenolic and flavonoid content of the bee pollen samples that were studied are given in Table 2. 

The botanical origin, as well as the geographical location from which the samples were collected, resulted in some significant differences in total phenolic content. For the TPC, the lowest value was recorded for BP1 samples, *Carduus* sp., with 2.9 ± 0.8 mg GAE/g, whereas the highest value was observed in the BP7 sample, corresponding with *Castanea* sp., which had a value of 35.8 ± 1.5 mg GAE/g. A high total phenolics value was also observed in monofloral *Castanea* bee pollen from Italy [24]. The BP6 and BP7 samples collected in Bragança had significantly (*p* < 0.05) higher values compared with samples from Nisa. BP4 and BP6, which presented the same botanical origin, *Rubus* sp., from different geographical areas, showed a total phenolic amount of 14.8 ± 0.3 and 19.3 ± 0.4 mg GAE/g, respectively, which are in accordance with the values previously described for these pollens [24]. The differences observed can be related to the fact that they are different sub-individuals of the same species and pollen origin. Moreover, the different environmental conditions [8], as well as the harvesting season, and the processing and storage conditions of the bee pollen, were previously described as having a significant impact on both the total phenolic compounds and nutritional values of the product [21]. Regarding the total flavonoid content, the pollen species from the Oleaceae family in BP2 showed a higher content, with a value of 4.8 ± 0.8 mg QE/g. This was followed by BP7 (3.1 ± 0.1 mg QE/g) and BP3 (2.5 ± 0.3 mg QE/g), respectively. Overall, both in terms of total phenolic and flavonoid content, Portuguese monofloral bee pollen samples from *Ligustrum*/*Olea* sp. and *Castanea* sp. exhibited a remarkable bioactive compound composition. These outputs are in line with the previously reported findings of flavonoid compounds in bee pollen, which found a dominant abundance of pollen types, such as *Castanea*, in the Northwest of Spain [14], as well as high TFC levels found in olive pollen (compared with palm pollen), which belongs to the Oleaceae family [22]. 

This indicates that natural bee pollen and other bee products are dominated by species such as *Castanea* sp., *Rubus* sp., or *Ligustrum*/*Olea* sp., and thus, they could be good candidates for use as foodstuffs, or to produce functional foods, due to their richness in bioactive compounds.

### 2.3. LC/DAD/ESI-MS^n^ Bioactive Compounds Analysis 

The specific bioactive composition of the monofloral bee pollen was successfully identified and quantified after extraction using the LC/DAD/ESI-MS^n^ chromatographic method, based on its molecular structure. The proposed compounds, with their retention time, maximum absorption wavelengths, and mass spectral data in negative ion mode, are summarized in Table 3. The interpretation of fragmentation pathways in the MS^n^ spectra of individual bioactive compounds was evaluated by comparing it with the results reported in the literature, and by assessing it alongside spectral information from the maximum UV.

The chromatographic profile allowed the identification of two major groups of compounds in all extracts, as follows: (i) flavonoids, mainly flavonols, and (ii) phenylamides. These were present in a total of seventy-two compounds (Table 3). A variety of different flavonol glycosides were identified in all extracts, with different degrees of substitution, mainly involving kaempferol, quercetin, isorhamnetin, herbacetin, myricetin, and laricitrin. The sugar moieties in the flavonoids were assigned to pentosides, hexosides, and deoxyhexosides. Additionally, acetyl- and malonyl glycosides that were linked to the flavonols were also found. As denoted in previous studies [6,10,12,23], these compounds are frequently identified in bee pollen samples with different botanical origins, and are responsible for the biological activities of bee pollen.

The total flavonoid compounds of samples BP1 and BP3 were very diverse, but the individual contribution of each compound was not higher than 15%. The same does not apply to sample BP7, wherein three isorhamnetin glycosides accounted for more than 62% of the flavonoid compounds. This flavonoid aglycone was specific to *Castanea* sp. pollen. For BP2, dominated by the *Ligustrum*/*Olea* sp. pollen species, quercetin-*O*-diglucoside (*m*/*z* 625) was detected at a high concentration, with a value of 4.88 ± 0.09 mg/g (65% of the total flavonoids). For BP4 and BP6, although collected in different geographical regions, the phenolic profile was similar, which indicates that they were of the same floral origin, *Rubus* sp. Even though some small variations in the individual quantities of each compound were observed between the samples, it is notable that the major compounds were the same. Additionally, this botanical origin seems very rich in methyl herbacetin glycosides. Conversely, in *Echium* sp. pollen, BP5, kaempferol glycosides are the major flavonoids, with kaempferol-3-*O*-rutinoside (*m*/*z* 593) found in the highest concentration, at 2.61 ± 0.03 mg/g. It is clear that bee pollen samples contain several phenolic compounds, but it seems that there is a specific flavonoid abundance which corresponds with the botanical origin.

LC-MS analysis also revealed the presence of another group of compounds that are common in bee pollen—phenylamides. Qiao et al. recently confirmed the richness of monofloral bee pollen from China by identifying 31 phenylamides [31]. These compounds are found in high amounts in the pollen structure of higher plants, both in mono- and di-cots [32]. Phenylamides are derivatives of polyamines and are considered plant-specific secondary metabolites with a variety of functional roles under stress conditions [33]. They play an important role, not only against biotic stress, but also against abiotic stress, as it is the defensive mechanism of phenylamides, which works against plant pathogens; this has been well summarized in previous studies [33,34]. These findings help explain the accumulation of phenylamides in high amounts, in various tissues of plants (including pollen grains). These low-molecular compounds are products of covalent bonding between carboxylic groups of hydroxycinnamic acids, and amine groups of aliphatic di- and polyamines or aromatic monoamines [32]. Caffeic, ferulic, and *p*-coumaric acids are the most common acidic moieties of phenylamides, whereas aliphatic polyamines, spermidine, and spermine are their dominant amine components [32].

The total amount of phenylamide compounds in the bee pollen samples varied widely, in the range of 0.3 to 25.6 mg/g (Table 3). Among the samples, the lowest phenylamide levels were observed for the Cistaceae and Boraginaceae families (samples BP3 and BP5), at 0.33 mg/g and 2.01 mg/g, respectively, whereas the maximum levels were found in samples BP4 and BP7, with values above 20 mg/g, which corresponded to *Rubus* sp. and *Castanea* sp. *N*^1^-feruloyl-*N*^5^, *N*^10^-dicaffeoylspermidin (isomer) (*m*/*z* 644), *N*^1^, *N*^5^, *N*^10^-tri-*p*-coumaroylspermidine (isomer) (*m*/*z* 582), diferuloyl coumaroyl spermidine (isomer), (*m*/*z* 642) and tetracoumaroyl spermine (isomer) (*m*/*z* 785) were detected at high concentrations, and are responsible for the high values of phenylamide observed in the BP4, BP6, and BP7 samples. The contribution of the phenylamides to the total concentration of bioactive compounds varied significantly, from 5% in *Cistaceae* sp., to 86% in *Castanea* sp. A previous work on bee pollen samples, collected from the northern region of Portugal, and dominated by *Crepis capillaris*, *Cytisus striatus*, and *Plantago* sp., reported that phenylamides constituted 86% to almost 100 % of the total bioactive compounds; high concentrations of phenylamide compounds such as *N*^1^-acetyl-*N*^5^, *N*^10^-di-*p*-coumaroylspermidine, *N*^1^, *N*^5^-di-*p*-coumaroyl-*N*^10^-caffeoylspermidine, *N*^1^, *N*^5^-di-*p*-coumaroyl-*N*^10^-caffeoylspermidine, tetracoumaroyl spermine, and its isomers were found [10]. More recently, a study on *Castanea* bee pollen from the north of the Iberian Peninsula described a phenolic profile resembling the one of sample BP7, where *N*^1^, *N*^5^, *N*^10^-tri-*p*-coumaroylspermidine, and *N*^1^, *N*^5^, *N*^10^-tricaffeoylspermidine were the most abundant compounds, with a concentration up to 9 mg/g and 12 mg/g, respectively [12]; this corresponds with the values obtained from the BP7 sample.

The wide variability in phenylamide concentrations between samples can be explained in two possible ways, as follows: first, there are significant differences between the total bioactive compounds of bee pollen samples, as they are dependent on plant origin; second, the phenylamide compounds accumulated in the vacuoles of the pollen, and in the bilayer wall (exine and intine), may have been affected by the extraction method in different ways, since the extraction yield of each compound can be influenced by the wall thicknesses, structure, and apertures of the pollen grain [8].

### 2.4. GC-MS Volatile Compounds Analysis

The SPME-GC-MS analysis in the current study aimed to identify the relationships between the volatiles at relative concentrations. The compound identification was performed using a commercial MS database (NIST 2011 Mass Spectral Library). A Linear Retention Index (LRI) was calculated for each detected component. For the calculation of the LRI indices, a mixture of n-alkanes (C7–C40) was used as a reference series. The quantification was obtained directly from the total number of ions in the chromatogram (Total Ion Chromatogram, TIC), and it was expressed as a relative percentage. 

A total number of 49 volatile compounds were identified in the bee pollen samples. The list of volatile compounds, along with the retention time (RT), relative concentration (R%), and LRI, are displayed in Table 4. 

The identified compounds cover different classes of organic structures, including terpenes (as thymol or eucalyptol), linear alcohols (as octanol or 1-hexanol), aldehydes (as nonanal or decanal), ketones (as jasmone), hydrocarbons (as nonadecane or hexadecane), esters (as ethyl decanoate or ethyl nonanoate), and fatty acids (as decanoic acid). 

Esters were detected in all the samples with different R%: BP1 contained less than 20%, whereas the ester composition of BP7 was more than 80%. Terpenes were only identified in the BP4, BP5, and BP6 samples. Yet, only the latter sample comprised compounds that did not belong to the abovementioned classes, at a level of more than 25%. *Castanea* sp., BP7, was composed of almost 69% nonanals, and a significant %R profile of alcohols; this could be associated with the nectar that the bees used to form the bee pollen pellets, or even the honeybee itself. BP1 was the only sample that presented fatty acids among all the pollen samples. Similar to our results, Kaškonienė et al. [35], Starowicz et al. [36], and Kaškonienė et al. [37] reported that hexanal, heptanal, 6-methyl-5-hepten-2-one, and nonanal compounds were frequently detected at higher concentrations in bee pollen samples from countries such as Lithuania, Poland, Spain, Latvia, and China. 

Most of the studies investigating the volatile composition of bee pollen have been conducted without referring to the botanical origin of the pollen, and therefore, they lack the ability to associate specific volatile compounds with their plant origin. Therefore, after identifying the botanical origins and volatile organic compounds of our monofloral bee pollen samples, we discuss whether there is a potential relationship between pollen type and compounds.

Although there are compounds, such as nonanal, hexanal, or methyl-5-hepten-2-one, which are abundantly common, in all samples, there are other compounds that are more specific. *Carduus* sp., sample BP1, showed some specific carbonyl derivatives, such as benzaldehyde, dodecanal, 3,5,5-trimethyl-3-cyclohexen-1-one, or fatty acids such as decanoic acid. Moreover, *Ligustrum*/*Olea* sp. pollen, sample BP2, revealed the presence of jasmone and 1,3,5-trimethoxy-benzene, whereas the terpenes, eucalyptol and linalool, appear distinctly in sample BP6, which is dominated by *Rubus* sp. It is relevant to note that although samples BP4 and BP6 are from the same genus, their volatile profile does not overlap, which may reflect the differences found between the species; this is understandable considering that the pollen was collected from distinct geographical origins. In sample BP3, from *Cistaceae* sp., the presence of unsaturated hydrocarbons, such as 7-hexadecene and octadecyne, seems to differ between pollen samples, but nerylacetone, which is only found in BP5, also stands out due to the high quantity found. For *Castanea* sp., the most specific compound is the alcohol, octanol, but it is important to note that the presence of nonanal is at least six times higher than in any other samples.

Previous studies concerning volatile and semi-volatile organic compounds, secreted by bee glands, reported (*E*)-citral as an orientation pheromone secreted by the worker bees’ Nasonov gland [38]. The same compound was detected in both BP4 and BP5 samples. Moreover, nonanal was detected in worker bees’ pheromone samples, as were octanol, hexanol, benzaldehyde, and caryophyllene, which were also reported in other studies [39,40].

### 2.5. Biological Activity

#### 2.5.1. Antioxidant Activity

Plants produce a broad spectrum of bioactive compounds, like phenolic acids, flavonoids, and anthocyanins; they contain one or more hydroxyl groups that are attached to either a phenyl ring or classes such as carotenoids. These compounds are important because they may impact several functions in humans, given that they act as antioxidants [41]. Phenolic compounds are unique oxygen radical scavengers because the electron reduction potential of phenolic radicals is lower than that of oxygen radicals, and they are usually less reactive than oxygen radicals [8,41].

Table 2 shows the results from the antioxidant assays of bee pollen samples. Accordingly, the highest DPPH^•^ radical scavenging activity was reached in the BP7 sample, with EC_50_: 0.07 ± 0.00 mg/mL, followed by BP2 (0.15 ± 0.00 mg/mL) and BP6 (0.20 ± 0.02 mg/mL). In the ABTS^•+^ test, the highest value was recorded in BP4, at 1.59 ± 0.15 mM Trolox/mg, followed by BP6, BP7, and BP2, at similar values. BP1 and BP5 samples were the lowest performers in both DPPH^•^ and ABTS^•+^ tests. It seems that samples with higher values for total phenolic and flavonoid compounds tend to show higher DPPH^•^ and ABTS^•+^ radical scavenging activity. This trend was evaluated via correlation analysis, and as is evident in Table 5, the total phenolic compounds were moderately correlated with DPPH^•^/ABTS^•+^ (r = 0.589/0.543), confirming the observation made in numerous other studies concerning bee pollen samples of various plant origins [10,16,25]. The moderate correlation with the antiradical assays can be related to other compounds present in bee pollen composition, as well as antioxidant activity, such as carotenoids [42]. 

The reducing power test was recorded in a range varying between a minimum of 0.02 ± 0.01 mg GAE/g (BP1 and BP4) and maximum of 0.09 ± 0.0 mg GAE/g (BP3) (Table 2). Surprisingly, the BP3 sample exhibited the highest reducing power activity, even though both the total phenolic/flavonoid and LC-MS results revealed that it exhibited a moderate total bioactive compound, as compared with the other samples. This was likely because of the diversity of the compounds in BP3’s composition, rather than its total bioactive compounds. As previously shown by Aylanc et al. [43], higher antioxidant activity may depend on the position of the functional group in the phenolic structure; the authors demonstrated that the H atoms of the attached hydroxyl groups (*o*–diphenol), at different locations of the rings (A, B, and C), as well as the double bonds of the benzene ring, and the double bond of the oxo functional group (–C=O), of some flavonoids provide compounds with a higher antioxidant capacity. Overall, considering the results of the different assays which predict antioxidant activity, the monofloral bee pollen samples from *Ligustrum*/*Olea* sp., *Rubus* sp., and *Castanea* sp. exhibited better results.

#### 2.5.2. Cytotoxic Activity

The biological activities of plant-based materials, including anti-tumour activity, are often correlated with the type and concentration of their phenolic compounds in their chemical composition [5,6]. From this perspective, each bee pollen sample was screened for their potential and in vitro cytotoxic activity against human cancer-derived cells, and a non-tumour cell line was used as a control. When evaluating cytotoxic activity, a lower 50% growth inhibition (GI_50_) value corresponds with higher cytotoxic activities, and values higher than 1000 μg/mL were considered to be inactive against the tested tumour cell lines. The GI of the cells was not significant (GI_50_ > 1000) for all bee pollen extracts against AGS, CaCo, HeLa, and NCI tumour cell lines (Table 6). 

Among the bee pollen factions, BP1, BP6, and BP7 exhibited cytotoxicity effects exclusively against the MCF-7 cell line—breast adenocarcinoma. The cytotoxic GI_50_ value of BP6 and BP7 against the MCF-7 tumour cell line was recorded as 814 ± 10.0 μg/mL and 746 ± 1.0 μg/mL, respectively. These values did not indicate any notable cytotoxic activity. However, BP1 showed effective cytotoxic activity against the same cell line with a value of 17 ± 1.1 μg/mL. Even though BP6 and BP7 showed a higher concentration profile in terms of bioactive compounds when compared with BP1, a lower cytotoxic activity could be attributed to the presence of some specific bioactive compounds in the BP1 chemical composition.

As shown in Table 3, the presence of various flavonoids (especially kaempferol, which was only determined to have a relatively high concentration in BP1) may be responsible for its anti-tumour activity. This was investigated previously by Lee et al. [44]. In this study, it has been demonstrated that kaempferol is a chemopreventive compound that significantly inhibited triclosan and bisphenol A-induced breast cancer cell proliferation and anti-apoptosis effects; this occurred via the regulation of pro- and anti-apoptosis, production of reactive oxygen species, and stress response-related genes. Moreover, in a literature review with a similar perspective, the researchers discussed the issue, and presented ample evidence to support the efficacy of flavonoids on breast cancer [45], which is in alignment with our results. Moreover, Chekuri et al. [46] showed that quercetin inhibited MCF-7 and MDA-MB-231 cells, as evaluated by apoptosis and cell cycle phase measurements. On the other hand, none of the tested bee pollen extracts showed toxic effects against a normal cell line (hFOB (human fetal osteoblastic)). These results indicate that monofloral bee pollen, dominated by *Carduus* sp., could be explored due to its notable activity against breast adenocarcinoma, due to the existence of some specific bioactive molecules, such as kaempferol and quercetin derivatives,

## 3. Materials and Methods 

### 3.1. Standards and Reagents 

Methanol, ethanol, acetonitrile, potassium ferrocyanide, and trichloroacetic acid were purchased from Fisher Scientific (Pittsburgh, PA, USA). Phenolic compounds standards (p-coumaric acid, quercetin, and chrysin), ABTS^•+^ [2,2′-azinobis-(3-ethyl-benzothiazoline-6-sulfonic acid)], Trolox (6-hydroxy-2,5,7,8-tetramethylchromane-2-carboxylic acid), and DPPH^•^ (2,2-diphenyl-1-picrylhydrazyl) were purchased from Sigma Chemical Co. (St Louis, MO, USA), and the kaempferol was purchased from Extrasynthese (Genay, France). Folin-Ciocalteu’s reagent and sodium chloride were purchased from Panreac Applichem (Barcelona, Spain). Iron (III) chloride, aluminium chloride, and naringenin were purchased from Acros Organics (Pittsburgh, PA, USA), and sodium carbonate anhydrous was purchased from Labkem (Barcelona, Spain). Water was treated in a Milli-Q water purification system (TGI pure system, Houston, TX, USA).

### 3.2. Sample Collection

Bee pollen samples were collected from the following two locations by local beekeepers during the spring/summer season in 2018: Nisa and Bragança, Portugal. After collection, the samples were cleaned of debris of wood and dead bee parts, and the pollen loads of each sample were manually separated by colour, shape, and size. Botanical origin was confirmed through palynological analysis. Samples were stored in the freezer (–20 °C), until time for further analysis.

### 3.3. Palynological Analysis 

Palynological analysis was performed in accordance with a previously described method [47]. Approximately 10 mL of distilled water was added to 1 g of bee pollen sample, and the mixture was vortexed vigorously. Then, a 100 μL aliquot was placed on a slide, and after drying, one drop of glycerine jelly was added for permanent preparation. Pollen grain identification was performed using an optical microscope. A reference collection from the botanical laboratory of the University of Vigo, Spain, and different pollen morphology guides were used for the identification of pollen types. The relative frequency of each pollen type was calculated by counting a minimum of 500 pollen grains per slide.

### 3.4. Phenolic Compounds Analysis

#### 3.4.1. Extraction Procedure

Phenolic compounds were extracted in accordance with a previously described procedure [12]. An aliquot of 2 g of fresh bee pollen sample was accurately ground and extracted with 15 mL of ethanol/water (70:30, *v*/*v*), at 70 °C, for 30 min, in a water bath under mechanical shaking conditions. The mixture was vacuum filtered, and the derived extract was stored at −20 °C until time for further analysis. 

#### 3.4.2. Total Phenolic and Flavonoid 

The total phenolic compounds of the hydroethanolic extracts of bee pollen samples were quantified spectrophotometrically, in accordance with the method reported by Falcão et al. [48]. Briefly, 0.5 mL of the extract (5 mg/mL) was mixed with Folin–Ciocalteu’s reagent (0.25 mL). After 3 min, 1 mL of 20% Na_2_CO_3_ was added to the mixture, and distilled water was added until the volume reached 5 mL. The solution was kept at 70 °C for 10 min and then cooled in the dark for 20 min. Thereafter, the mixture was centrifuged for 10 min at 5000× *g*, and the absorbance was measured to be 760 nm (Analytikijena 200–2004 spectrophotometer from Analytik Jena, Jena, Germany). The total phenolic content was expressed in mg of gallic acid equivalents per g of fresh bee pollen (mg GAE/g).

The aluminium chloride method was used to determine the total number of flavonoid compounds [10]. An aliquot of 0.2 mL of sample solution (5 mg/mL) was mixed with 0.2 mL of AlCl_3_ solution (2% aluminium chloride in acetic acid/methanol, 5/95, *v*/*v*). Then, 2.8 mL of methanol with 5% glacial acetic acid was added. The mixture was then incubated in the dark for 30 min, and the absorbance was measured at 415 nm using a spectrophotometer (Analytikijena 200–2004 spectrophotometer from Analytik Jena, Jena, Germany). The total number of flavonoid compounds were expressed in mg of quercetin equivalents per g of fresh bee pollen (mg QE/g).

#### 3.4.3. LC/DAD/ESI-MS^n^ Analysis

For the analysis, 20 mg of bee pollen extract was dissolved in 2 mL of 80% ethanol/water. The final solution was filtered through a 0.22 μm membrane and kept in the freezer at −20 °C, until time for analysis. A Dionex UltiMate 3000 ultra-pressure liquid chromatography instrument, connected to a diode array and attached to a mass detector, was used for LC/DAD/ESI-MS^n^ analyses (Thermo Fisher Scientific, San Jose, CA, USA). LC was conducted in a Macherey–Nagel Nucleosil C18 column (250 mm × 4 mm id; particles diameter of 5 mm, end-capped) and its temperature was kept constant at 30 °C. The conditions for the liquid chromatography process were based on a previous work [12], the flow rate was 1 mL/min and the injection volume was 10 μL. The final spectra data were accumulated in the wavelength interval of 190–600 nm. The LTQ XL linear ion trap mass spectrometer (Thermo Fisher Scientific, San Jose, CA, USA) equipped with an ESI source was set in the negative ion mode. The ESI conditions were in accordance with those described in previously reported studies [12]. Mass spectra were acquired via full range acquisition covering 100–1000 *m*/*z*. For the fragmentation study, a data-dependent scan was performed by deploying a collision-induced dissociation (CID). The normalized collision energy of the CID cell was set at 35 (arbitrary units). Data acquisition was conducted using the Xcalibur^®^ data system (Thermo Scientific, San Jose, CA, USA).

The elucidation of the phenolic compounds was achieved by comparing their chromatographic behaviour, UV spectra, and MS information, with those of reference compounds. When standards were not available, the structural information was confirmed via a combination of UV data and MS fragmentation patterns that were previously reported in the literature. Quantification was achieved using calibration curves for *p*-coumaric acid (0.00925–0.4 mg/mL; y = 2.06 × 10^7^x − 3.5 × 10^5^; R^2^ = 0.973), kaempferol (0.037–1.6 mg/mL; y = 4.27 × 10^6^x + 1.98 × 10^5^; R^2^ = 0.983), chrysin (0.0185–0.8 mg/mL; y = 1.20 × 10^7^x − 5.83 × 10^4^; R^2^ = 0.999), quercetin (0.037–1.6 mg/mL; y = 3.9 × 10^6^x + 4.65 × 10^5^; R^2^ = 0.937), and naringenin (0.0185–0.8 mg/mL; y = 7.85 × 10^6^x − 3.04 × 10^5^; R^2^ = 0.978). When the standard was not available, the compound quantification was expressed as the equivalent of the structurally closest compound. The results were expressed in mg/g of fresh pollen. 

### 3.5. Volatile Analysis

#### 3.5.1. Solid Phase Microextraction (SPME)

The extraction procedure was conducted in accordance with the previously described method [9]. Approximately, 2.5 g of ground fresh bee pollen was mixed with 2.5 mL of a 30% sodium chloride solution in a glass bottle until homogeneous. The vial was sealed with a predrilled septum and placed in a thermostatic bath at 50 °C. Headspace sampling was conducted using a manual SPME holder equipped with a 65 µm polydimethylsiloxane/divinylbenzene (PDMS/DVB) StableFlex fibre (Supelco, Bellefonte, PA, USA). The volatile bee pollen compounds were sampled by inserting the fibre through the septum and exposing it to the headspace for 60 min with continuous stirring. The fibre was then retracted and transferred to the injector port of the gas chromatograph where the compounds were desorbed for 5 min.

#### 3.5.2. GC-MS Analysis

For the volatile analysis, a previously described method was used [9]. The GC-MS unit consisted of a Perkin Elmer system (GC Clarus^®^ 580 GC module and Clarus^®^ SQ 8 S MS module) and gas chromatograph equipped with a DB-5 MS fused-silica column (30 m × 0.25 mm i.d., film thickness 0.25 μm; J & W Scientific, Inc., Folsom, CA, USA), and the unit was interfaced with a Perkin–Elmer Turbomass mass spectrometer (software version 6.1, Perkin Elmer, Shelton, CT, USA). The SPME fibre was desorbed at 250 °C for 5 min. The oven temperature was programmed at 40–170 °C, at 3 °C/min, then subsequently at 25 °C/min up to 290 °C, followed by isothermal for 15 min. The transfer line temperature was 280 °C, the ion source temperature was 230 °C, and the carrier gas and helium were adjusted to a linear velocity of 40 cm/s^−1^. The ionization energy was 70 eV, the scan range was 40–300 u, and the scan time was 1 s. The compound identifications were based on a comparison of the obtained spectra with those of the NIST mass spectral library, and they were confirmed using linear retention indices, which were determined using the retention times of an n-alkane (C_7_–C_36_) (Supelco, Bellefonte, PA, USA) mixture that was analysed under identical conditions. Moreover, they were compared against published data, and when possible, with commercial standard compounds. Quantitation (average value for three replicates per sample) was conducted using relative values obtained directly from the peak total ion current (TIC).

### 3.6. Biological Activity

#### 3.6.1. Antioxidant Activity

Three different assays were used to assess the in vitro antioxidant potential of bee pollen, as follows: the DPPH^•^ and ABTS^•+^ radical scavenging assays, and the reducing power assay. The DPPH^•^ free radical scavenging activity of bee pollen was performed in accordance with the method described by Tomás et al. [15], with some modifications. An aliquot comprising 0.15 mL of the pollen extract solutions, with concentrations ranging from 0.034 to 0.5 mg/mL, was mixed with 0.15 mL of DPPH^•^ (0.024 mg/mL), and the absorbance was read at 515 nm using an ELX800 Microplate Reader (Bio-Tek Instruments, Inc., Winooski, VT, USA). The percentage of radical inhibition was calculated using the following equation:% Inhibition = [(A_DPPH_ − A_Sample_)⁄(A_DPPH_) × 100](1)

The amount of antioxidant needed to reduce the initial DPPH^•^ concentration by 50% (EC_50_) was determined by plotting the inhibition percentage against the extract concentration.

Reducing the power analysis, which measures the reduction of the ferric ion (Fe^3+^)-ligand complex to the ferrous (Fe^2+^) complex, was performed in accordance with the method described in Falcão et al. [43]. The bee pollen extract (5 mg/mL) was mixed with a sodium phosphate buffer (pH = 6.6, 1.25 mL) and 1.25 mL of potassium ferricyanide (1%), and the mixture was incubated at 50 °C for 20 min. Then, trichloroacetic acid (10%, 1.25 mL) was added. After centrifuging at 3000× *g* for 10 min, a volume of 1.25 mL was removed from the top solution to a new tube, and it was combined with 1.25 mL of water and 0.25 mL of 0.1% ferric chloride. The absorbance of the final solution was then read as 700 nm. The results were expressed in milligrams of gallic acid equivalent per gram of sample (mg GAE/g).

The ABTS^•+^ [2,2′-azinobis-(3-ethyl-benzothiazoline-6-sulfonic acid)] assay was conducted to determine whether the bee pollen samples could scavenge the ABTS radical cation, using Trolox as the standard; the assay was performed in accordance with the previously described method, with modifications [49]. A stock ABTS^•+^ solution was prepared from 7 mM ABTS^•+^ and 2.45 mM sodium persulfate in deionized water. The working ABTS^•+^ solution was diluted with distilled water to obtain an absorbance of 0.70 (± 0.02) at 734 nm. Bee pollen extract (5mg/mL, 40 µL) was added to the diluted ABTS^•+^ solution (960 µL) and mixed immediately. The mixture was incubated for 10 min in the dark, and the absorbance was determined at 734 nm. The percentage of inhibition was calculated using the following formula:% Inhibition = ((1 − A_Sample_))/A_Control_) × 100(2)

A_Sample_ was the absorbance of the ABTS^•+^ solution containing the sample, and A_control_ was the absorbance of ABTS^•+^ solution without the sample. The Trolox equivalent antioxidant capacity (TEAC) of the bee pollen samples (mM Trolox/mg bee pollen extract) was calculated using the calibration curve, as follows:TEAC (mM Trolox/mg BP extract) = ((% Inhibition Sample-b)/a × (aliquot volume (mL))/(bee pollen weight (mg))(3)
where a and b are the slope and intercept of the calibration curve.

#### 3.6.2. Cytotoxic Activity

To evaluate the cytotoxic activity of the bee pollen extract with the sulforhodamine B (SRB) colorimetric assay, 5 human tumour cell lines were used, as follows: AGS (stomach gastric adenocarcinoma), CaCo-2 (epithelial colorectal adenocarcinoma), HeLa (cervical 25 carcinoma), MCF-7 (breast adenocarcinoma), and NCI-H460 (non-small cell lung cancer); hFOB (human Fetal Osteoblastic) was also used, and it is a non-tumour cell line. The treatment solution was prepared from a 20 mg/mL hydroethanolic bee pollen extract, which was freeze-dried and then diluted to various concentrations (125 μg/mL to 2000 μg/mL).

The cell lines’ subcultures were performed in an RPMI-1640 medium enriched with 2 mM glutamine, 100 U/mL penicillin, 100 μg/mL streptomycin, and 10% FBS, and they were kept in a humidified air incubator containing 5% CO_2_ at 37 °C. After 24 h of incubation, the attached cells were then treated with different extract concentrations and incubated again for 48 h. Afterwards, the adherent cells were fixed with cold trichloroacetic acid (TCA 10%, 100 μL) and incubated at 4 °C for 1 h. Subsequently, the cells were washed with deionized water and dried. An SRB solution (SRB 0.1% in 1% acetic acid, 100 μL) was added to each well plate and incubated for 30 min at room temperature. The unbound SRB was removed with 1% acetic acid and the plates were air-dried. The bound SRB was solubilized with Tris (10 mM, 200 μL). To measure the absorbance at 540 nm, an ELX800 microplate reader (Bio-Tek Instruments, Inc.; Winooski, VT, USA) was used [27]. Elipticine was used as a positive control, and the results were expressed as GI_50_ values in μg/mL (sample concentration that inhibited 50% of the net cell growth).

### 3.7. Statistical Analysis

All analyses were performed in triplicate, and the results were denoted as a mean ± standard deviation (SD). The obtained data were analysed using GraphPad Prism version 9.3 (San Diego, CA, USA). The one-way analysis of variance, followed by Tukey’s multiple comparison test, were conducted to determine statistical significance. *p* < 0.05 was considered to be significant. Moreover, Pearson’s correlation coefficients were calculated to ascertain the relationship between the tested parameters.

## 4. Conclusions

Bee pollen is a bee product which has high potential for use as a human food supplement, and it may induce health benefits. It is evident, given the results in the literature, that those benefits are closely related to botanical origin, therefore, it is relevant that studies concerning its composition and biological activities are focused on monofloral pollen. In this study, we identified and quantified the phenolic composition and volatile compounds of seven Portuguese monofloral bee pollens, and we assessed their potential biological activities, such as their antiradical and anti-tumour activities. The findings confirmed that the plant origin of bee pollen has a major effect on the composition and diversity of compounds. The sample of *Castanea* sp. origin exhibited the highest DPPH^•^ and significant ABTS^•+^ radical scavenging and reducing power activity, with the highest content in terms of total phenolic compounds and phenylamide compounds. On the other hand, the bee pollen extracts did not reveal notable anti-tumour activity, except against MCF-7 (breast adenocarcinoma), whereas the sample BP1, dominated by *Carduus* sp, showed significant anti-tumour action, without revealing toxicity on non-tumour cells. Overall, bee pollen of *Carduus* sp., *Rubus* sp., and *Castanea* sp. origin could be a promising natural source of bioactive compounds and biological activities for the food and pharmaceutical industries.

## Figures and Tables

**Figure 1 molecules-28-07601-f001:**
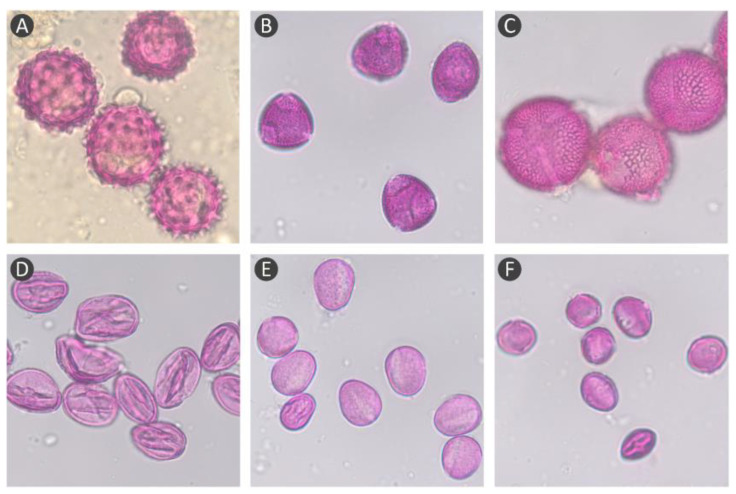
Microscopic view of the major pollen species seen in bee pollen samples. *Carduus* sp. (**A**); *Olea* sp. (**B**); *Cistaceae* sp. (**C**); *Rubus* sp. (**D**); *Echium* sp. (**E**); and *Castanea* sp. (**F**).

**Table 1 molecules-28-07601-t001:** Codes, colour, and botanical origin of bee pollen samples.

Origin	Sample Code	Visual Colour	Family	Predominant Pollen Type (%)
Nisa, PT	BP1	Beige	Asteraceae	*Carduus* sp. (˃80%)
	BP2	Yellow	Oleaceae	*Ligustrum*/*Olea* sp. (˃80%)
	BP3	Orange	Cistaceae	*Cistaceae* (100%)
	BP4	Light green	Rosaceae	*Rubus* sp. (100%)
	BP5	Purple	Boraginaceae	*Echium* sp. (100%)
Bragança, PT	BP6	Soft green	Rosaceae	*Rubus* sp. (100%)
	BP7	Dark yellow	Fagaceae	*Castanea* sp. (100%)
Classification				Monofloral

PT: Portugal, BP: bee pollen.

**Table 2 molecules-28-07601-t002:** Total phenolic (TPC), flavonoid compounds (TFC), and antioxidant activity of bee pollen.

Sample	TPC (mg GAE/g)	TFC (mg QE/g)	DPPH^•^ (EC_50_: mg/mL)	ABTS^•+^ (mM Trolox/mg)	Reducing Power (mg GAE/g)
BP1	2.9 ± 0.8 ^f^	0.7 ± 0.1 ^e^	0.38 ± 0.00 ^b^	0.98 ± 0.02 ^e^	0.02 ± 0.00 ^e^
BP2	16.0 ± 0.8 ^c^	4.8 ± 0.8 ^a^	0.15 ± 0.00 ^d^	1.33 ± 0.02 ^b^	0.03 ± 0.00 ^d^
BP3	6.8 ± 1.0 ^e^	2.5 ± 0.3 ^c^	0.25 ± 0.01 ^bc^	1.22 ± 0.15 ^c^	0.09 ± 0.01 ^a^
BP4	14.8 ± 0.3 ^d^	1.2 ± 0.2 ^d^	0.37 ± 0.00 ^b^	1.59 ± 0.15 ^a^	0.02 ± 0.01 ^e^
BP5	6.8 ± 1.1 ^e^	2.3 ± 0.3 ^c^	0.75 ± 0.04 ^a^	1.11 ± 0.05 ^d^	0.03 ± 0.00 ^d^
BP6	19.3 ± 0.4 ^b^	0.7 ± 0.0 ^e^	0.20 ± 0.02 ^cd^	1.35 ± 0.05 ^b^	0.04 ± 0.01 ^c^
BP7	35.8 ± 1.5 ^a^	3.1 ± 0.1 ^b^	0.07 ± 0.00 ^e^	1.34 ± 0.16 ^b^	0.05 ± 0.02 ^b^

BP: bee pollen. GAE: gallic acid equivalent, QE: quercetin equivalent. EC_50_: 50% efficient concentration. Different letters (^a–f^) in each column indicate significant (*p* < 0.05) differences between samples.

**Table 3 molecules-28-07601-t003:** Identification and quantification of the phenolic and phenylamide compounds of the different monofloral Portuguese bee pollen, obtained using LC/DAD/ESI-MS^n^.

t_R_ (min)	λ_max_ (nm)	[M − H]^−^ *m*/*z*	MS^n^ (% Base Peak)	Proposed Compound	BP1	BP2	BP3	BP4	BP5	BP6	BP7
**6.7**	259, 356	625	317	Myricetin-3-*O*-rutinoside ^a,c^	nd	nd	0.69 ± 0.02	nd	nd	nd	nd
**7.5**	257, 353	625	301 (100), 300 (90), 445 (82), 271 (15)	Quercetin-*O*-diglucoside ^a,c^	0.21 ± 0.00	4.88 ± 0.09	nd	nd	nd	nd	nd
**8.2**	258, 356	711	667	Myricetin-*O*-malonyl-rutinoside ^a,d^	nd	nd	0.19 ± 0.02	nd	nd	nd	nd
**8.5**	271, 325, 353sh	639	459 (100), 315 (91), 300 (34)	Methyl herbacetin-*O*-dihexoside ^a,e^	0.24 ± 0.02	nd	nd	4.23 ± 0.04	nd	2.38 ± 0.05	nd
**9.2**	258, 356	639	330 (100), 331 (47), 315 (21)	Laricitrin-*O*-hexosyl-deoxyhexoside ^a,f^	nd	nd	0.22 ± 0.01	nd	nd	nd	nd
**9.3**	265, 349	609	429 (100), 285 (98), 284 (40)	Kaempferol-*O*-dihexoside ^a,c^	0.20 ± 0.01	nd	nd	nd	nd	nd	nd
**9.6**	256, 355	609	300 (100), 301 (44)	Quercetin-3-*O*-rutinoside ^a,c^	nd	0.27 ± 0.01	0.41 ± 0.02	nd	nd	nd	nd
**10.3**	265, 349	609	429 (100), 285 (98), 284 (40)	Kaempferol-*O*-dihexoside (isomer) ^a,c,g^	0.16 ± 0.02	0.63 ± 0.06	nd	nd	nd	nd	nd
**10.5**	259, 355	565	521	Myricetin-*O*-malonyl hexoside ^a,c^	nd	nd	0.25 ± 0.01	nd	nd	nd	nd
**10.7**	271, 331, 358sh	725	681	Methyl herbacetin-malonyl-dihexoside ^a^	0.21 ± 0.02	nd	nd	2.05 ± 0.04	nd	2.07 ± 0.07	nd
**11.0**	266, 348	593	447 (100), 431 (48), 285 (8)	Kaempferol-*O*-hexosyl-*O*-deoxyhexoside ^a,c^	nd	nd	0.32 ± 0.02	nd	nd	nd	nd
**11.6**	254, 354	609	315	Isorhamnetin-*O*-pentosyl-hexoside ^a,c,k^	0.23 ± 0.00	nd	nd	nd	nd	nd	nd
**11.8**	256, 355	695	651	Quercetin-*O*-malonyl deoxyhexosyl-hexoside ^a,c^	nd	nd	0.24 ± 0.01	nd	nd	nd	nd
**12.1**	267, 347	593	284 (100), 285 (69)	Kaempferol-3-*O*-rutinoside ^a,e,h^	0.24 ± 0.01	0.46 ± 0.01	0.32 ± 0.02	0.21 ± 0.01	2.61 ± 0.03	nd	0.36 ± 0.00
**12.3**	254, 354	623	314 (100), 315 (85), 459 (80)	Isorhamnetin-3-*O*-hexosyl-deoxyhexoside ^a,h,k^	nd	nd	0.16 ± 0.00	nd	nd	nd	1.01 ± 0.01
**12.6**	256, 354	463	301	Quercetin-3-*O*-glucoside ^a,h^	nd	0.33 ± 0.01	0.18 ± 0.00	nd	nd	nd	nd
**12.6**	254, 354	609	315	Isorhamnetin-*O*-pentosyl-hexoside ^a,c,k^	nd	nd	nd	nd	nd	nd	0.62 ± 0.00
**12.6**	272, 327, 352sh	709	665	Methyl herbacetin-*O*-malonyl-hexosyl-deoxyhexoside ^a^	nd	nd	nd	0.21 ± 0.01	nd	0.22 ± 0.02	nd
**12.8**	253, 353	709	519	Isorhamnetin-*O*-acetyl-hexoside ^a,e^	nd	nd	0.16 ± 0.00	nd	nd	nd	nd
**13.0**	254, 354	609	315	Isorhamnetin-*O*-pentosyl-hexoside ^a,c,k^	nd	nd	nd	nd	nd	nd	0.67 ± 0.01
**13.0**	255, 354	695	651	Isorhamnetin-*O*-malonyl pentosyl-hexoside ^a,c^	0.46 ± 0.01	nd	nd	nd	0.15 ± 00	nd	nd
**13.3**	256, 353	549	505	Quercetin-*O*-malonyl hexoside ^a,c^	nd	nd	0.97 ± 0.01	nd	nd	nd	nd
**13.4**	266, 348	679	635	Kaempferol-*O*-malonyl-hexosyl-deoxyhesoxide ^a,d^	0.19 ± 0.01	0.28 ± 0.01	nd	nd	1.24 ± 0.01	nd	0.18 ± 0.00
**13.5**	272, 327, 352sh	477	315 (100), 462 (50), 300 (15)	Methyl herbacetin-*O*-hexoside ^a,d^		nd	nd	0.19 ± 0.01	nd	0.18 ± 0.02	nd
**13.5**	253, 256	579	535	Laricitrin-*O*-malonylhexoside ^a^	nd	nd	0.17 ± 0.00	nd	nd	nd	nd
**13.6**	254, 354	623	315	Isorhamnetin-3-*O*-hexosyl-deoxyhexoside (isómer) ^a,h,k^	nd	nd	nd	nd	nd	nd	0.32 ± 0.00
**13.9**	254, 353	549	505	Quercetin-*O*-malonyl hexoside (isomer) ^a,c^	nd	nd	0.17 ± 0.01	nd	nd	nd	nd
**14.1**	263, 347	447	285 (100), 284 (90)	Kaempferol-*O*-hexoside ^a,e^	nd	nd	0.16 ± 0.00	nd	0.17 ± 0.00	nd	nd
**14.3**	253.351	477	314 (100), 315 (53)	Isorhamnetin-3-*O*-glucoside ^a,c^	0.19 ± 0.01	nd	0.16 ± 0.00	nd	nd	nd	0.54 ± 0.02
**14.5**	299, 308	436	316	di-*p*-coumaroylspermidine ^a,i^	nd	0.16 ± 0.00	nd	nd	nd	nd	nd
**15.0**	265, 340	417	284 (100), 285 (48)	Kaempferol-*O*-pentoside ^a,j^	0.17 ± 0.00	nd	nd	nd	nd	nd	nd
**15.0**	295, 315	630	468 (100), 494 (86), 358 (7)	*N*^1^*, N*^5^*, N*^10^-tricaffeoylspermidine ^a,e^	nd	0.33 ± 0.02	nd	nd	nd	nd	0.39 ± 0.01
**15.0**	266, 344	679	635	Kaempferol-*O*-malonyl-hexosyl-deoxyhexoside (isómer) ^a,d^	nd	nd	0.25 ± 0.01	nd	nd	nd	nd
**15.4**	255, 355	563	519	Isorhamnetin-*O*-malonyl-hexoside (isomer) ^a,d^	0.39 ± 0.03	nd	0.29 ± 0.00	nd	nd	nd	nd
**15.8**	295, 316	630	468 (100), 494 (85), 495 (20)	*N*^10^*, N*^5^*, N*^10^-tricaffeoylspermidine ^a,e^	nd	nd	nd	nd	nd	nd	1.84 ± 0.03
**16.3**	256, 370	447	301	Quercetin-3-*O*-rhamnoside ^a,h^	nd	nd	0.85 ± 0.01	nd	nd	nd	
**16.2**	298, 318	630	468 (100), 494 (86), 358 (7)	*N*^1^*, N*^5^*, N*^10^-tricaffeoylspermidine (isómer) ^a,e^	nd	nd	nd	nd	nd	nd	0.26 ± 0.01
**16.8**	299.319	630	468 (100), 494 (84), 358 (7)	*N*^1^*, N*^5^*, N*^10^-tricaffeoylspermidine (isómer) ^a,e^	nd	nd	nd	nd	nd	nd	8.83 ± 0.01
**17.6**	293, 315	644	508 (100), 468 (11), 482 (11)	*N*^1^-feruloyl-*N*^5^*, N*^10^-dicaffeoylspermidin ^a,e^	nd	nd	nd	nd	nd	nd	0.34± 0.04
**18.0**	296, 315	614	478 (100), 452 (69), 494 (25), 358 (20)	*N*^1^-*p*-coumaroyl-*N*^5^*, N*^10^-dicaffeoylspermidine ^a,e^	nd	0.24 ± 0.00	nd	nd	nd	nd	0.41 ± 0.05
**18.1**	299, 310	598	462 (100), 452 (45), 478 (40)	*N*^1^*, N*^5^-di-*p*-coumaroyl-*N*^10^-caffeoylspermidine ^a,e^	nd	nd	nd	0.23 ± 0.00	nd	0.22 ± 0.03	nd
**18.3**	296, 315	614	478 (100), 452 (69), 494 (25), 358 (20)	*N*^1^-*p*-coumaroyl-*N*^5^*, N*^10^-dicaffeoylspermidine (isomer) ^a,e^	nd	0.19 ± 0.02	nd	nd	nd	nd	0.30 ± 0.05
**18.7**	296, 315	614	478 (100), 452 (68), 468 (20), 342(5)	*N*^1^-*p*-coumaroyl-*N*^5^*, N*^10^-dicaffeoylspermidine (isomer) ^a,e^	nd	nd	nd	nd	nd	nd	0.72 ± 0.02
**19.0**	299, 310	598	462 (100), 478 (41), 452 (39),	*N*^1^*, N*^5^-di-*p*-coumaroyl-*N*^10^-caffeoylspermidine (isomer) ^a,e^	nd	nd	nd	0.42 ± 0.05	nd	0.44 ± 0.01	nd
**19.3**	298, 308	614	478 (100), 452 (78), 494 (24), 358 (18)	*N*^1^-*p*-coumaroyl-*N*^5^*, N*^10^-dicaffeoylspermidine (isomer) ^a,e^	0.15 ± 0.02	0.57 ± 0.03	nd	0.35 ± 0.05	nd	0.40 ± 0.01	2.00 ± 0.03
**19.5**	297, 319	644	508 (100), 468 (11), 482 (11)	*N*^1^-feruloyl-*N*^5^*, N*^10^-dicaffeoylspermidine (isomer) ^a,e^	nd	nd	nd	nd	nd	nd	0.83 ± 0.04
**20.0**	297, 319	644	508 (100), 482 (80), 468 (4)	*N*^1^-feruloyl-*N*^5^*, N*^10^-dicaffeoylspermidine (isomer) ^a,e^	nd	0.14 ± 0.00	nd	nd	nd	nd	1.33 ± 0.03
**20.0**	271, 356	623	315 (100), 300 (42), 477 (14)	Methylherbacetin-3-*O*-rutinoside ^a,f^	0.26 ± 0.00	nd	nd	nd	nd	nd	nd
**20.3**	299, 310	598	462 (100), 478 (46),452 (46), 342 (14)	*N*^1^, *N*^5^-di-*p*-coumaroyl-*N*^10^-caffeoylspermidine (isomer) ^a,e^	nd	0.16 ± 0.01	nd	0.84 ± 0.02	nd	0.33 ± 0.01	nd
**20.6**	264, 341	431	285	Kaempferol-3-*O*-rhamnoside ^a,h^	nd	nd	0.17 ± 0.00	nd	nd	nd	nd
**21.2**	290, 309	582	462	*N*^1^*, N*^5^*, N*^10^-tri-*p*-coumaroylspermidine ^a,e,^	nd	nd	nd	1.22 ± 0.01	nd	0.41 ± 0.01	0.66 ± 0.01
**21.3**	268, 347	285	285 (100), 257 (13), 151 (20)	Kaempferol ^a,b^	0.27 ± 0.00	nd	nd	nd	nd	nd	nd
**21.5**	273	612	492	Feruloyl dicoumaroyl spermidine (isomer) ^a,i^	nd	nd	nd	nd	0.17 ± 0.02	nd	nd
**22.2**	299, 310	598	462 (100), 478 (39), 452 (34), 342 (14)	*N*^1^*, N*^5^-di-*p*-coumaroyl-*N*^10^-caffeoylspermidine (isomer) ^a,e^	0.23 ± 0.00	0.44 ± 0.01	nd	2.37 ± 0.01	nd	0.65 ± 0.01	2.07 ± 0.08
**22.8**	295, 310	582	462 (100), 436 (9), 342 (7)	*N*^1^*, N*^5^*, N*^10^-tri-*p*-coumaroylspermidine (isomer) ^a,e^	nd	nd	nd	2.34 ± 0.03	nd	0.54 ± 0.01	0.37 ± 0.00
**23.1**	291, 310	612	492	Feruloyl dicoumaroyl spermidine (isomer) ^a,i^	nd	nd	nd	nd	0.22 ± 0.06	nd	nd
**24.0**	291, 310	642	522 (100), 492 (78), 506 (57)466 (16),	Diferuloyl coumaroyl spermidine (isomer) ^a,i^	nd	nd	nd	nd	0.16 ± 0.02	nd	nd
**24.1**	295, 310	582	462 (100), 436 (9), 342 (6)	*N*^1^*, N*^5^*, N*^10^-tri-*p*-coumaroylspermidine (isomer) ^a,c^	0.14 ± 0.00	0.23 ± 0.00	nd	3.63 ± 0.01	nd	0.79 ± 0.01	1.04 ± 0.01
**24.5**	293, 310	612	492	Feruloyl dicoumaroyl spermidine (isomer) ^a,i^	nd	nd	nd	nd	0.21 ± 0.00	nd	nd
**25.0**	295, 310	583	462 (100), 436 (9), 342 (7)	*N*^1^*, N*^5^*, N*^10^-tri-*p*-coumaroylspermidine (isomer) ^a,c^	nd	nd	nd	1.37 ± 0.08	nd	0.40 ± 0.01	0.88 ± 0.01
**25.4**	291, 310	642	522 (100), 492 (78), 506 (57)466 (16),	Diferuloyl coumaroyl spermidine (isomer) ^a,i^	nd	nd	nd	nd	0.18 ± 0.03	nd	nd
**25.4**	367	271	151 (100), 176 (10)	Naringenin ^a,b^	nd	0.66 ± 0.05	nd	nd	nd	nd	nd
**26.2**	294, 310	612	492	Feruloyl dicoumaroyl spermidine (isomer) ^a,i^	nd	nd	nd	nd	0.21 ± 0.05	nd	nd
**26.5**	295, 310	582	462 (100), 436 (10), 342 (7)	*N*^1^*, N*^5^*, N*^10^-tri-*p*-coumaroylspermidine (isomer) ^a,c^	0.54 ± 0.01	0.72 ± 0.03	0.18 ± 0.02	12.2 ± 0.02	0.17 ± 0.00	7.13 ± 0.09	0.87 ± 0.01
**27.0**	270	785	665	Tetracoumaroyl spermine (isomer) ^a,d,e^	0.50 ± 0.01	nd	nd	nd	nd	nd	nd
**27.6**	298, 310	612	492	Feruloyl dicoumaroyl spermidine (isomer) ^a,d,e,i^	nd	0.19 ± 0.00	nd	0.44 ± 0.03	0.28 ± 0.00	0.19 ± 0.01	nd
**28.6**	276	785	665	Tetracoumaroyl spermine (isomer) ^a,d,e^	0.77 ± 0.01	0.20 ± 0.00	0.15 ± 0.01	nd	0.23 ± 0.00	nd	nd
**29.3**	280, 310sh	785	665	Tetracoumaroyl spermine (isomer) ^a,d,e^	0.17 ± 0.02	nd	nd	nd	nd	nd	nd
**30.6**	291	785	665	Tetracoumaroyl spermine (isomer) ^a,d,e^	0.25 ± 0.01	nd	nd	nd	nd	nd	nd
**32.8**	294, 310	785	665	Tetracoumaroyl spermine (isomer) ^a,d,e^	0.21 ± 0.01	nd	nd	nd	nd	nd	nd
**34.0**	299, 310	785	665	Tetracoumaroyl spermine (isomer) ^a,d,e^	0.99 ± 0.01	2.11 ± 0.01	nd	0.19 ± 0.00	0.18 ± 0.04	nd	nd
				Total flavonoids (mg/g)	3.16 ± 0.05	7.51 ± 0.10	6.33 ± 0.05	6.89 ± 0.05	4.17 ± 0.03	4.85 ± 0.01	3.70 ± 0.02
				Total phenylamides (mg/g)	4.21 ± 0.04	5.68 ± 0.05	0.33 ± 0.02	25.6 ± 0.10	2.01 ± 0.10	11.5 ± 0.10	23.1 ± 0.10

Confirmed with: ^a^ MS^n^ fragmentation; ^b^ standard; References: ^c^ [9]; ^d^ [25].; ^e^ [10]; ^f^ [26]; ^g^ [27]; ^h^ [28]; ^i^ [29]; ^j^ [30]; ^k^ [12]. nd: not detected. The values are expressed as mg of each compound per g of bee pollen.

**Table 4 molecules-28-07601-t004:** Bee pollen volatile composition, obtained via GC-MS.

RT	Compound	LRI	BP1	BP2	BP3	BP4	BP5	BP6	BP7
**3.1**	Hexanal	768	2.8 ± 1.8	2.3 ± 1.1	3.8 ± 0.2	16.1 ± 8.1	7.3 ± 1.1	8.6 ± 1.9	0.1 ± 0.9
**4.2**	2-hexenal	830	1.1 ± 0.9	nd	0.8 ± 0.1	nd	nd	nd	nd
**5.0**	1-hexanol	849	nd	nd	nd	nd	nd	4.1 ± 1.3	1.8 ± 0.1
**5.4**	Heptanal	883	0.5 ± 0.3	1.2 ± 0.1	0.4 ± 0.4	nd	nd	3.0 ± 0.7	1.4 ± 0.1
**6.1**	Methyl hexanoate	909	0.3 ± 0.3	2.4 ± 1.3	0.9 ± 0.5	nd	nd	nd	nd
**7.3**	(*E*,*E*)-2,4-heptadien-6-ynal	943	nd	nd	nd	nd	nd	nd	0.4 ± 0.4
**7.3**	Benzaldehyde	944	1.2 ± 0.5	nd	nd	nd	nd	nd	nd
**8.1**	1-octen-3-ol	965	nd	nd	nd	nd	1.8 ± 0.1	nd	0.3 ± 0.1
**8.2**	6-Methyl-5-hepten-2-one	970	3.8 ± 1.4	3.3 ± 0.4	7.7 ± 0.8	3.9 ± 0.9	3.9 ± 0.3	7.6 ± 6.6	1.8 ± 0.2
**8.6**	2,4-heptadienal	998	6.5 ± 2.1	2.8 ± 0.1	0.5 ± 0.4	nd	nd	2.4 ± 0.7	nd
**8.9**	Octanal	988	nd	nd	2.1 ± 1.4	4.3 ± 1.2	nd	6.0 ± 1.9	13.9 ± 1.2
**9.2**	2,4-heptadienal (isomer)	999	nd	nd	7.6 ± 1.4	2.4 ± 0.8	nd	nd	nd
**9.9**	Eucalyptol	1016	nd	nd	nd	nd	nd	2.1 ± 0.3	nd
**10.4**	3,5,5-trimethyl-3-cyclohexen-1-one	1026	0.9 ± 0.1	nd	nd	nd	nd	nd	nd
**11.7**	3,5-octadien-2-one	1057	10.4 ± 3.4	4.5 ± 0.8	nd	12.5 ± 0.4	18.1 ± 1.1	5.6 ± 2.6	nd
**11.8**	Octanol	1059	nd	nd	nd	nd	nd	nd	8.1 ± 1.3
**12.6**	3,5-octadien-2-one (isomer)	1079	nd	nd	7.7 ± 0.8	5.6 ± 0.8	7.9 ± 0.1	nd	nd
**13.0**	Linalool	1086	nd	nd	nd	nd	nd	5.6 ± 0.9	nd
**13.2**	Nonanal	1091	2.7 ± 1.9	2.5 ± 2.4	11.6 ± 0.8	7.8 ± 2.6	7.3 ± 0.2	0.2 ± 0.2	66.9 ± 1.8
**13.4**	Phenylethylalcohol	1097	nd	nd	nd	nd	1.2 ± 0.4	nd	nd
**13.7**	Isophorone	1104	nd	nd	nd	nd	0.7 ± 0.6	nd	nd
**14.1**	Methyl octanoate	1111	1.7 ± 0.1	5.7 ± 4.2	4.8 ± 1.1	nd	nd	nd	nd
**15.6**	Nonenal	1144	nd	nd	nd	nd	1.8 ± 0.2	nd	nd
**16.2**	Nonanol	1159	nd	nd	nd	nd	nd	4.5 ± 1.7	2.6 ± 0.1
**17.3**	Ethyl octanoate	1183	5.4 ± 0.7	8.1 ± 3.7	nd	nd	nd	nd	nd
**17.7**	Decanal	1191	0.7 ± 0.2	nd	nd	nd	1.4 ± 0.1	6.5 ± 1.1	nd
**19.0**	Cis-verbenol	1219	nd	nd	nd	1.9 ± 0.7	1.5 ± 0.7	nd	nd
**19.1**	Linalyl anthranilate	1223	nd	nd	nd	nd	nd	2.4 ± 0.4	nd
**20.3**	Citral	1248	nd	nd	nd	3.3 ± 0.4	2.2 ± 0.5	nd	nd
**21.4**	Thymol	1273	nd	nd	nd	1.9 ± 0.3	1.6 ± 0.2	nd	nd
**22.6**	3-(3-methyl-1-butenyl)cyclohexene	1299	0.7 ± 0.7	nd	nd	nd	nd	nd	nd
**22.9**	Methyl decanoate	1307	1.4 ± 0.3	7.4 ± 2.2	nd	nd	nd	nd	nd
**25.3**	N-Decanoic acid	1360	1.6 ± 0.8	nd	nd	nd	nd	nd	nd
**25.7**	Jasmone	1370	nd	4.8 ± 1.4	nd	nd	nd	nd	nd
**25.9**	Ethyl hexadecanoate	1377	nd	nd	nd	0.9 ± 0.1	nd	nd	nd
**26.0**	Ethyl decanoate	1377	nd	2.5 ± 0.4	4.8 ± 0.2	nd	nd	nd	nd
**26.4**	1,3,5-trimethoxy-benzene	1386	nd	5.5 ± 1.3	nd	nd	nd	nd	nd
**26.5**	Dodecanal	1388	1.1 ± 0.3	nd	nd	nd	nd	nd	nd
**28.0**	Nerylacetone	1423	nd	nd	16.5 ± 0.7	nd	3.1 ± 0.1	nd	nd
**28.0**	6,10-dimethyl-5,9-undecadien-2-one	1425	6.4 ± 1.5	7.4 ± 8.1	nd	nd	nd	15.9 ± 0.2	nd
**29.6**	2,6,10-trimethyltetradecane	1462	nd	nd	nd	nd	0.6 ± 0.4	nd	nd
**30.1**	Hexadecane	1474	nd	nd	nd	6.9 ± 1.8	5.9 ± 0.7	nd	nd
**31.2**	Methyl dodecanoate	1500	4.4 ± 0.5	10.2 ± 5.5	nd	nd	nd	nd	nd
**33.2**	7-hexadecene	1551	nd	nd	4.4 ± 0.7	nd	nd	nd	nd
**36.3**	9,12,15-octadecatrienal	1642	nd	14.4 ± 0.2	nd	0.4 ± 0.9	2.1 ± 0.1	nd	nd
**36.4**	9,12,15-octadecatrien-1-ol	1634	2.7 ± 0.9	nd	nd	nd	nd	nd	nd
**36.5**	Octadecyne	1636	nd	nd	4.3 ± 0.3	nd	nd	nd	nd
**37.0**	8-heptadecene	1649	36.5 ± 1.2	nd	nd	21.7 ± 2.2	21.7 ± 1.0	10.9 ± 5.1	nd
**37.8**	Nonadecane	1671	6.4 ± 3.0	nd	22.1 ± 1.3	10.4 ± 0.7	9.1 ± 0.8	14.4 ± 2.1	nd

RT: Retention time (min); LRI: Linear Retention Index determined using a DB-5 MS fused silica column, relative to a series of n-alkanes (C7–C36); nd: non detected The values are expressed as relative percentages (%).

**Table 5 molecules-28-07601-t005:** Correlation analysis between total phenolic compounds and antioxidant capacity.

Assay	r	Significance Level (*p* Value)
TPC–DPPH^•^	0.589	0.164
TPC–ABTS^•+^	0.543	0.208
TPC–Reducing power	0.071	0.879
TFC–DPPH^•^	0.247	0.593
TFC–ABTS^•+^	0.106	0.822
TFC–Reducing power	0.216	0.642

Twenty-one paired average samples from each test were used in the comparison. r value represents Pearson’s linear correlation value. The level of significance was expressed as *p* < 0.05.

**Table 6 molecules-28-07601-t006:** Cytotoxicity activity (GI_50_ values, µg/mL) of BP samples (*n* = 3, mean ± SD).

Cell Lines	GI_50_, µg/mL						
	BP1	BP2	BP3	BP4	BP5	BP6	BP7
AGS	>1000	>1000	>1000	>1000	>1000	>1000	>1000
CaCo-2	>1000	>1000	>1000	>1000	>1000	>1000	>1000
HeLa	>1000	>1000	>1000	>1000	>1000	>1000	>1000
MCF-7	17 ± 1	>1000	>1000	>1000	>1000	814 ± 10	746 ± 1
NCI-H460	>1000	>1000	>1000	>1000	>1000	>1000	>1000
hFOB	>1000	>1000	>1000	>1000	>1000	>1000	>1000

## Data Availability

Data are contained within the article.

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
