# Peer review of "Differentiating between Monofloral Portuguese Bee Pollens Using Phenolic and Volatile Profiles and Their Impact on Bioactive Properties"

_molecules, 2023, doi:10.3390/molecules28227601_

Round 1

Reviewer 1 Report

Comments and Suggestions for Authors

The study differentiated pollen origins using their phenolic and volatile profiles, identifying a total of 72 compounds with the LC/DAD/ESI-MS method and 49 compounds via GC-MS, including phenolics and various volatile compounds. Among the samples, Castanae sp. pollen showed the strongest DPPH radical scavenging activity, while Rubus sp. and Cistaceae sp. demonstrated high antioxidant activity; only Carduus sp. displayed significant cytotoxic potential against MCF-7.

The authors should indicate the significance of the study and emphasize its novelty at the end of the introduction section. The difference of this study from the other literature studies should be clearly indicated.

The title of Table 4 should be revised. The second sentence should be at the bottom of Table.

The materials and methods section are enough to explain the details

Results and discussion part is also sufficient. However, I do not prefer to see the explanation of the cytotoxic activity results as indicated in Line 338-339 “These results indicate that monofloral bee pollen, dominated by Carduus sp., could be explored due to its notable activity against breast adenocarcinoma, due to the existence of some specific bioactive molecules”.

The authors should at least suggest some compound groups looking at the previous literature studies. What kind of compounds play active roles in the cytotoxic activity breast adenocarcinoma?

Author Response

The manuscript is now revised taking into consideration the proposals of the reviewers. Below you can find the specific actions regarding each reviewer's comments.
We would like to thank the reviewers for their valuable comments, suggestions, and contributions to this study.

Reviewer #1:
1. The authors should indicate the significance of the study and emphasize its novelty at the end of the introduction section. The difference of this study from the other literature studies should be clearly indicated.
Our response: We improved the introduction section according to the reviewer's suggestion.
2. The title of Table 4 should be revised. The second sentence should be at the bottom of Table.
Our response: Changes were made accordingly to the reviewer's suggestion.
3. The materials and methods section are enough to explain the details.
Results and discussion part is also sufficient. However, I do not prefer to see the explanation of the cytotoxic activity results as indicated in Line 338-339 “These results indicate that monofloral bee pollen, dominated by Carduus sp., could be explored due to its notable activity against breast adenocarcinoma, due to the existence of some specific bioactive molecules”.
The authors should at least suggest some compound groups looking at the previous literature studies. What kind of compounds play active roles in the cytotoxic activity breast adenocarcinoma?
Our response: Considering the reviewer's suggestion we improved the discussion by adding information on flavonoids linked with the cytotoxic activity, from previous studies. Besides, we concluded our discussion by highlighting the compounds most probably linked with the observed effect on BP1.

Reviewer 2 Report

Comments and Suggestions for Authors

The manuscript entitled "Differentiation of monofloral Portuguese bee pollen through the phenolic and volatile profile and its impact in the bioactive properties" is an interesting attempt to gain more information about monofloral bee pollen origin from Portugal.

The authors evaluate the phenolic and volatile profile of seven samples of bee pollen and investigate their antioxidant and cytotoxic activity.

The biological activity of the compounds may not have been exceptional but the overall composition of the manuscript is good, with a very nice introduction to the topic. The quality of the figures and tables is also good. The paper is written in a clear and understanding manner and the results are properly presented. The compounds are sufficiently characterized from the chemical point of view.

Overall I have find this manuscript suitable for publication in “Molecules” in current form.

Author Response

Reviewer #2:

The manuscript entitled "Differentiation of monofloral Portuguese bee pollen through the phenolic and volatile profile and its impact in the bioactive properties" is an interesting attempt to gain more information about monofloral bee pollen origin from Portugal.

The authors evaluate the phenolic and volatile profile of seven samples of bee pollen and investigate their antioxidant and cytotoxic activity.

The biological activity of the compounds may not have been exceptional but the overall composition of the manuscript is good, with a very nice introduction to the topic. The quality of the figures and tables is also good. The paper is written in a clear and understanding manner and the results are properly presented. The compounds are sufficiently characterized from the chemical point of view.

Overall I have find this manuscript suitable for publication in “Molecules” in current form.I have found some minor details in the ms that need to be revised, hopefully improving the present work's quality.

Our response: Thank you the reviewer for their valuable comments. Since no specific recommendations were given by the reviewer we made a general revision of the ms data to improve the outputs.

Reviewer 3 Report

Comments and Suggestions for Authors

The manuscript is a study aiming to differentiate the monofloral bee pollen originating in Portugal. The analysis is based on the phenolic and volatile compounds found in bee pollen and their antioxidant and cytotoxic activity. The authors performed the palynological analysis of collected samples to identify the pollen grains. Then, they obtained hydroalcoholic extracts with phenolic compounds to determine the total phenolic and flavonoid content. A liquid chromatograph (LC) coupled with a diode array detector (DAD), electrospray ionization (ESI) and mass spectrometry (MS) was used to identify and quantify phenolic and phenolamide compounds such as aldehydes, esters, ketones, hydrocarbons, and terpenes. Volatile compounds were separated using solid phase microextraction (SPMS) and analysed using a gas chromatograph coupled with mass spectrometry (GC-MS). The antioxidant activity was determined using DPPH and ABTS radical scavenging assays and reducing power assay. Finally, the cytotoxic activity was evaluated using five human tumour cell lines.

The manuscript is written well and structured according to the Instructions for the authors, although the reverse of the sections „Materials and Methods” and „Results and Discussion” is atypical.

The results are presented in six tables, discussed and compared with the literature. The results and conclusions are important for food and pharmaceuticals since the study shows 72 compounds identified and quantified, indicating the species with the highest antioxidant activity and cytotoxic potential.

Comments on the Quality of English Language

The manuscript is well written using a clear English language. However, several minor language errors have been identified. A few examples are presented below.

L3 Correct the title by replacing „in” with „on”.

L30 Choose a synonym of „exhibited” to avoid repetition. Suggestion: „showed”.

L45 It appears that „essentially” may be unnecessary in this sentence. Consider removing it.

L46 The pronoun „own” may be redundant here. Consider removing it.

L67 Insert a comma before „which” and remove the article „the” before „geographical”.

The authors have to verify the whole manuscript.

Author Response

The manuscript is now revised taking into consideration the proposals of the reviewers. Below you can find the specific actions regarding each reviewer's comments.

We would like to thank the reviewers for their valuable comments, suggestions, and contributions to this study.

Reviewer #3:

The manuscript is well written using a clear English language. However, several minor language errors have been identified. A few examples are presented below.

  1. L3 Correct the title by replacing „in” with „on”.

Our response: Changes were made according to the reviewer's suggestion.

  1. L30 Choose a synonym of „exhibited” to avoid repetition. Suggestion: „showed”.

Our response: Changes were made according to the reviewer's suggestion.

  1. L45 It appears that „essentially” may be unnecessary in this sentence. Consider removing it.

Our response: Changes were made according to the reviewer's suggestion.

  1. L46 The pronoun „own” may be redundant here. Consider removing it.

Our response: Although we understand the point raised by the reviewer we think the word highlights the origin of the secretions.

  1. L67 Insert a comma before „which” and remove the article „the” before „geographical”.

Our response: Changes were made according to the reviewer's suggestion.

  1. The authors have to verify the whole manuscript.

Our response: The entire manuscript was revised for minor improvements, according to the reviewer's suggestion.

Reviewer 4 Report

Comments and Suggestions for Authors

I have some suggestions and requests towards authors to be considered.

Major suggestions:

1. Main objection and request is related to chemical terminology used. Namely, term "phenolamides" must be corrected with "phenylamides" to be chemically punctual. Namely, phenol group, in this case is substituent on amide group due to its priority in Organic chemistry (amide is one five the most prior functional groups in Organic chemistry. Please revise a whole document carefully.

2. Please revise carefully a whole text related to the Latin names. Namely, all names for plant species and genera must be given in Italic while plant family names should not be given.

3. Please specify for all quantification data are you express obtained results based on fresh or dry weight of sample? It is quite important and quite different to interpret further results.

4. DPPH is a radical and must be labeled uniformly and always in this way. In the same time, ABTS is radical cation and must be labeled uniformly and always in this way. Please revise a whole document carefully.

Minor corrections: All suggestions are given below with an appropriate Line number(s) from text in order to facilitate tracking.

Lines 46-48: Suggest to authors to exclude this sentence about pollen as gametes, etc. It is too general and well known so there is no need for repeating once again.

Line 54: Strongly suggest that here authors also specify what pollen sample can be marked as monofloral according to prof. Campos suggestion.

Line 77: I totally agree that phenylamides were overlooked in bee-collected pollen analysis but situation is drastically changed in the last couple of years. In that sense, I can not agree that we here have only one, suggested, reference no. 10. In that sense, I would ask to authors to perform additional literature search and to include the most novel researches about phenylamides identification and quantification in bee-collected pollen samples with different botanical origin. There are several published articles from this and previous year.

Line 95: Is it possible that authors include picture of examined pollen samples? I think it would be very informative and useful for future readers.

Line 103: "content" here. It is uncountable noun, except in case of book contents.

Lines 1177-180: Here also should be specified that presence of these derivatives in pollen is quite expected since polyamines, parenting molecules, are strong fighters in pollen against oxidative stress, both biotic and abiotic. There are several review articles about this topic.

Line 184: Strongly suggest to replace "our" with "examined" here. Personal pronouns, should not be used in scientific language.

Line 247: "showed" in paste tense here? Check/correct.

Line 264: Is this "t study" here typo? Check/correct.

Line 277: Suggest to replace "special" with "important" here. It seems to me much more appropriate in this context.

Lines 278-279: I think that here authors should also mention carotenoids as important ROS eliminators regardless the fact that you did not examine carotenoids. Pollen is usually excellent source of carotenoids also so its contribution should not be overlooked.

Line 286: Please replace this "worst" with more appropriate term. Also, I think it should be here "in both assays"? Check/correct.

Line 292: Related to correlation analysis interestingly TFC was not in the line with antioxidant activity. This also should be mentioned. Again, maybe carotenoids also contribute antioxidant activity in examined pollen samples?

Line 303: Put letter "o" in name of "o-diphenol" in Italic.

Line 313: Put "in vitro" term in Italic here.

Line 315: "corresponded" in past tense here? Check/correct.

Line 325: I do not understand on what pollen "fraction" authors referring here? I did not realize from all previous discussion that you have prepared any special fraction from samples. Please clarify.

Line 384: Put letter "g" for centrifuge here in Italic.

Line 417: Put letter "p" in name of "p-coumaric acid" in Italic.

Line 443: typos- delete . after "C" in units for GC-MS analysis program.

Line 455: Put "in vitro" term in Italic here.

Line 467: Please specify here that this is "Ferric Reducing Power" assay. There are also "cuppric" assay for instance.

Line 470: Put letter "g" for centrifuge here in Italic.

Comments on the Quality of English Language

Observed minor possible errors related to English are suggested to authors in section for authors.

Author Response

The manuscript is now revised taking into consideration the proposals of the reviewers. Below you can find the specific actions regarding each reviewer's comments.

We would like to thank the reviewers for their valuable comments, suggestions, and contributions to this study.

Reviewer #4:

I have some suggestions and requests towards authors to be considered.

Major suggestions:

  1. Main objection and request is related to chemical terminology used. Namely, term "phenolamides" must be corrected with "phenylamides" to be chemically punctual. Namely, phenol group, in this case is substituent on amide group due to its priority in Organic chemistry (amide is one five the most prior functional groups in Organic chemistry. Please revise a whole document carefully.

Our response: The changes were made in the manuscript according to the reviewer's suggestion.

  1. Please revise carefully a whole text related to the Latin names. Namely, all names for plant species and genera must be given in Italic while plant family names should not be given.

Our response: A full revision was made in the manuscript according to the reviewer's suggestion.

  1. Please specify for all quantification data are you express obtained results based on fresh or dry weight of sample? It is quite important and quite different to interpret further results.

Our response: Additional information was given in the section of Materials and Methods to clarify the question raised by the reviewer.

  1. DPPH is a radical and must be labeled uniformly and always in this way. In the same time, ABTS is radical cation and must be labeled uniformly and always in this way. Please revise a whole document carefully.

Our response: A full revision was made in the manuscript according to the reviewer's suggestion.

Minor corrections: All suggestions are given below with an appropriate Line number(s) from text in order to facilitate tracking.

  1. Lines 46-48: Suggest to authors to exclude this sentence about pollen as gametes, etc. It is too general and well known so there is no need for repeating once again.

Our response: The changes were made in the manuscript according to the reviewer's suggestion.

  1. Line 54: Strongly suggest that here authors also specify what pollen sample can be marked as monofloral according to prof. Campos suggestion.

Our response: Changes were made according to the reviewer's suggestion.

  1. Line 77: I totally agree that phenylamides were overlooked in bee-collected pollen analysis but situation is drastically changed in the last couple of years. In that sense, I can not agree that we here have only one, suggested, reference no. 10. In that sense, I would ask to authors to perform additional literature search and to include the most novel researches about phenylamides identification and quantification in bee-collected pollen samples with different botanical origin. There are several published articles from this and previous year.

Our response: Considering the reviewer's suggestion, a few studies indicating the presence of phenylamide compounds, for various pollen species, were included in the section Results and Discussion.

  1. Line 95: Is it possible that authors include picture of examined pollen samples? I think it would be very informative and useful for future readers.

Our response:  A figure with pictures of the examined pollen samples was added according to the reviewer’s suggestion.

  1. Line 103: "content" here. It is uncountable noun, except in case of book contents.

Our response: The word ''content'' refers to the quantification value for total phenol and total flavonoids. To avoid confusion, we removed it from the title.

  1. Lines 1177-180: Here also should be specified that the presence of these derivatives in pollen is quite expected since polyamines, parenting molecules, are strong fighters in pollen against oxidative stress, both biotic and abiotic. There are several review articles about this topic.

Our response: Considering the reviewer's suggestion, a paragraph on the relationship between the presence of phenylamides and stress conditions was added to the "LC/DAD/ESI-MSn Bioactive Compounds Analysis" section.

  1. Line 184: Strongly suggest to replace "our" with "examined" here. Personal pronouns, should not be used in scientific language.

Our response: A correction was made to the text.

  1. Line 247: "showed" in paste tense here? Check/correct.

Our response: A correction was made to the text.

  1. Line 264: Is this "t study" here typo? Check/correct.

Our response: A correction was made to the text.

  1. Line 277: Suggest to replace "special" with "important" here. It seems to me much more appropriate in this context.

Our response: A correction was made to the text.

  1. Lines 278-279: I think that here authors should also mention carotenoids as important ROS eliminators regardless the fact that you did not examine carotenoids. Pollen is usually excellent source of carotenoids also so its contribution should not be overlooked.

Our response: We include the reference to carotenoids within the chemical compounds with bioactive importance, nevertheless we did not emphasize its specificities since the evaluation of this class within our samples was out of the scope.

  1. Line 286: Please replace this "worst" with more appropriate term. Also, I think it should be here "in both assays"? Check/correct.

Our response: Changes were made according to the reviewer's suggestion.

  1. Line 292: Related to correlation analysis interestingly TFC was not in the line with antioxidant activity. This also should be mentioned. Again, maybe carotenoids also contribute antioxidant activity in examined pollen samples?

Our response: Corrections were made according to the suggestion of the reviewer.

  1. Line 303: Put letter "o" in name of "o-diphenol" in Italic.

Our response: Changes were made according to the reviewer's suggestion.

  1. Line 313: Put "in vitro" term in Italic here.

Our response: Changes were made according to the reviewer's suggestion.

  1. Line 315: "corresponded" in past tense here? Check/correct.

Our response: It was accepted as simple tense and was left in the same form in the manuscript.

  1. Line 325: I do not understand on what pollen "fraction" authors referring here? I did not realize from all previous discussion that you have prepared any special fraction from samples. Please clarify.

Our response: The text was corrected since the meaning was referring to pollen samples and not pollen fractions.

  1. Line 384: Put letter "g" for centrifuge here in Italic.

Our response: Changes were made according to the reviewer's suggestion.

  1. Line 417: Put letter "p" in name of "p-coumaric acid" in Italic.

Our response: Changes were made according to the reviewer's suggestion.

  1. Line 443: typos- delete . after "C" in units for GC-MS analysis program.

Our response: Changes were made according to the reviewer's suggestion..

  1. Line 455: Put "in vitro" term in Italic here.

Our response: Changes were made according to the reviewer's suggestion.

  1. Line 467: Please specify here that this is "Ferric Reducing Power" assay. There are also "cuppric" assay for instance.

Our response: Changes were made according to the reviewer's suggestion.

  1. Line 470: Put letter "g" for centrifuge here in Italic.

Our response: Changes were made according to the reviewer's suggestion.